# Uniaxial and Coaxial Electrospinning for Tailoring Jussara Pulp Nanofibers

**DOI:** 10.3390/molecules26051206

**Published:** 2021-02-24

**Authors:** Sergiana dos P. Ramos, Michele A. Giaconia, Marcelo Assis, Paula C. Jimenez, Tatiana M. Mazzo, Elson Longo, Veridiana V. De Rosso, Anna R. C. Braga

**Affiliations:** 1Department of Biosciences, Universidade Federal de São Paulo (UNIFESP), Silva Jardim Street, 136, Vila Mathias, Santos, SP 11015-020, Brazil; sergiana.passos@unifesp.br (S.d.P.R.); michele.giaconia@unifesp.br (M.A.G.); veridiana.rosso@unifesp.br (V.V.D.R.); 2Department of Chemical, CDMF/LIEC (UFSCar) P.O. Box 676, São Carlos, SP 13560-970, Brazil; marcelostassis@gmail.com (M.A.); elson.liec@gmail.com (E.L.); 3Institute of Marine Sciences, Universidade Federal de São Paulo (UNIFESP), P.O. Box, Santos, SP 11070-100, Brazil; pcjimenez@unifesp.br (P.C.J.); tatimazzo@gmail.com (T.M.M.); 4Department of Chemical Engineering, Universidade Federal de São Paulo (UNIFESP), Campus Diadema, Diadema, SP 09972-270, Brazil

**Keywords:** nanostructures, response surface methodology, polyethylene oxide, anthocyanins, bioactive compounds, *Euterpe edulis*

## Abstract

Jussara pulp (*Euterpe edulis* Mart.) is rich in bioactive compounds known to be protective mediators against several diseases. In this context, nevertheless, anthocyanins, the most abundant natural pigment in jussara, are sensitive to temperature, pH, oxygen, and light conditions, leading to instability during food storage or digestion, and, thus jeopardizing the antioxidant proprieties retained by these flavonoids and limiting industrial application of the pulp. The production of nanostructures, from synthetic and natural polymers, containing natural matrices rich in bioactive compounds, has been widely studied, providing satisfactory results in the conservation and maintenance of the stability of these compounds. The current work aimed to compare uniaxial and coaxial electrospinning operation modes to produce core-shell jussara pulp nanofibers (NFs). Additionally, the parameters employed in the electrospinning processes were optimize using response surface methodology in an attempt to solve stability issues for the bioactive compounds. The best experimental conditions provided NFs with diameters ranging between 110.0 ± 47 and 121.1 ± 54 nm. Moreover, the coaxial setup improved jussara pulp NF formation, while further allowing greater integrity of NFs structures.

## 1. Introduction

In food sciences, anthocyanins should be examined to an extent beyond their most obvious traits as natural pigments, for these are remarkable bioactive compounds present in fruits and vegetables capable of delivering favorable effects to human health [1,2,3]. The most frequent property described is related to their antioxidant activity, since these natural pigments act as free radical scavengers and are frequently linked to prevention of chronic diseases [4,5,6]. Nevertheless, despite their beneficial properties, the efficiency of anthocyanins depends on their stability and bioavailability in the sourced matrix [2], and these bioactive compounds are susceptible to pH, temperature, oxygen and light conditions [7].

The fruit of *Euterpe edulis* Mart., a palm tree from the Atlantic Forest, is commonly known as jussara. This fruit presents high contents of anthocyanin and is very similar to the açai berry in both its sensory and nutritional characteristics. Studies carried out by our research group revealed that jussara, in addition to having colorant power, has several health-related benefits [8,9,10]. Therefore, jussara can be considered an excellent food matrix to explore anthocyanin properties, including the industrial application as naturals pigments in processed foodstuff and beverages.

The growing demand for healthier and more sustainable food has led to the search for new ingredients. In this context, the formation of micro- and nanostructures with bioactive compounds produces additional health benefits, protecting the functional molecules from adverse physicochemical conditions inherent to storage or consumption/digestion [11]. In this context, nanostructures are generally classified within a diameter size range between 1 and 100 nm [12,13].

Several encapsulation methods and techniques have already been applied to use phenolic compounds attempting to overcome the unstable nature of this food constituent, such as spray drying, freeze-drying extrusion and electrospinning, among others [8,14,15]. Electrospinning is a simple technique that does not require specific reagents or solutions nor extreme temperatures. It is, indeed, a top-down process to produce nanostructures, particularly core-shell nanofibers (NFs) composed by bioactive compounds, which further allows to scale-up the process [16,17,18]. This process occurs using a high electrical field between the polymeric solution and the collector, in which there is a potential gradient. When a sufficiently high voltage is reached, a jet of polymer solution will erupt from a droplet of the mixture. Over this distance, the solution is evaporated, forming the NF. This technique has been employed to produce continuous fibers from a wide variety of materials [19,20].

The electrospinning process admits two operational modes. In the uniaxial method, the compound of interest and the carrier material are mixed previously and ejected through the application of the electric field using a single needle. In contrast, in the coaxial configuration, the polymer solution and the compound are expelled simultaneously from two different needles. The polymer represents the outer shell of the NF, and the compound is placed in the core portion of the structure [7,21,22]. Both setups have their own advantages and drawbacks with respect to one another: the coaxial configuration permits a greater control of the flow system, while the uniaxial technique is simpler to operate. The coaxial method of electrospinning is a modification of the classic (uniaxial) procedure capable of producing micro/nano fibers with core-shell structures; however, in this method, it is not possible to form micro/nano particles. Both methods depend on a series of system control parameters. Applied voltage, collector distance and flow rate are fundamental for determining the diameter size of the structures formed.

Several polymers can be used in food products; however, they must earn a GRAS (Generally Recognized as Safe) grade from the Food and Drug Administration (FDA), which is the case of the water-soluble synthetic polymer polyethylene oxide (PEO) [23]. Still, anthocyanins and other compounds are arranged within the fruit matrix, which ensures their stabilization; therefore, the use of whole jussara pulp is best indicated to produce core-shell NFs since the extraction process could lead to degradation of these biomolecules and decrease of their beneficial effects [24]. Furthermore, to produce smaller diameters and more uniform and smooth nanostructures through electrospinning, the polymeric solutions may be added with salt, such as NaCl. This commonly used strategy allows an increased number of ions in the solution and more elongation forces are imposed on the jet under the electrical field [25,26].

Pondering these considerations, we hypothesize, herein, that changes in the electrospinning parameters for production of NFs containing jussara pulp result in distinct structures regarding, minimally, shape and size. Therefore, the present work aims to maximize the means for tailoring jussara pulp NFs through electrospinning using response surface methodology and employing both operation setups—uniaxial and coaxial—thus establishing an alternative but efficient process for stabilizing bioactive compounds in foodstuffs. Therefore, the present work aims to compare two methods of producing NFs by electrospinning—uniaxial and coaxial—for preserving and maintaining the stability of jussara pulp, using surface methodology to maximize the means for this.

## 2. Results and Discussion

### 2.1. Electrospinning Process

Eleven electrospun solutions (Table 1) produced fibers with various diameters and characteristics, therefore generating a set amenable for evaluation. Morphological structures of the NFs are shown in Figure 1. The working parameters employed herein were based on previous work [21].

FE-SEM images of electrospinning solutions (Figure 1) display the manufactured fibers. Among samples 1, 2, 4, 5, 6, and 7 (Figure 1b,c,e–h), only slight variations, considering their morphology, size and distribution of the fibers, can be perceived. Sample 3 (Figure 1d), in turn, displays higher heterogeneity of the fibers. Figure 1j–l provide images from each of the three replicates generated for samples 9–11, in which the elevated degree of intrasample similarity is indicative of a high reproducibility of the process employed herein for obtaining the NFs. Sample 8 (Figure 1i) is singled out considering the presence of large spaces between fibers and occurrence of regions where fibers appear to be aggregated. None of the samples show homogeneous fibers in terms of size or orientation. The diameters of the respective fibers were measured from the FE-SEM images and data is presented in Table 1. Diameter measurements were used to verify the effects of PEO and NaCl concentration over jussara pulp NFs obtained through this experimental design.

### 2.2. Experimental Design: Response Surface Methodology (RSM)

The results obtained for NFs diameter (nm) in the CCRD regarding the two variables studied—PEO and NaCl concentration—are shown in Table 1. The parameters, which were not statistically significant, were incorporated into the lack of fit for calculation of R^2^ and F ratio. The NFs diameter varied from 135.0 to 4268.0 nm, among which sample 2 yielded the lowest value acquired—a response to PEO concentration at level +1 (7.7%) and NaCl concentration at level −1 (0.36%).

The main effects and interactions estimated for jussara pulp NFs diameter were reported in Appendix A. The dependent and independent variables were fitted to a model of second order and examined in terms of goodness of fit. Therefore, ANOVA was used to assess the adequacy of the fitted model (Appendix A).

The R^2^ value provided a measure of how much the experimental factors and their interactions could explain the variability in the observed response values (Appendix A). A good model (values above 0.9 are considered very good) explains most of the variation in the response. As closer as the R^2^ value is to 1.00, the stronger the model and the better the response predictions [27]. In this study, the R^2^ value of 0.94 and the superior performance of the F ratio obtained, returns a 95% confidence in the model.

The central points provided additional degrees of freedom for error estimation, which increases power when testing the significance of effects. In our work, the pure error (Appendix A) was deficient, indicating good reproducibility of the experimental design. Thus, the coded model was considered predictive, and it can be used to generate the response surface for the ANOVA effects on the electrospun NFs diameter containing jussara pulp according to the model Equation (1).
Diameter (nm) = 1554.53 + 627.02 (X_2_) + 651.65 (X_2_)^2^ + 1364 (X_1_·X_2_)(1)

An overview of Equation (1), in which X_1_ and X_2_ correspond to PEO and NaCl concentration, respectively, indicates that the diameter of the NFs is a first-order function of the NaCl concentration, while the interaction between PEO and NaCl is a second-order function for NaCl concentration. Equation (1) shows the equation model fitted according to the regression analysis of the values of diameter experimentally determined.

The model for NFs was used to construct the response surfaces seen in Figure 2 in order to understand the interactions between the independent variables tested (PEO and NaCl concentration) and the optimum concentration of each component required for maximization of NF production. Figure 2a,b shows two work regions that would result in NFs with smaller diameters: the first, using low PEO concentrations and high NaCl amounts; and the second, when PEO is high and NaCl is low. Sample 2, the only condition of the experimental design presenting nanosized diameter fibers, fits the second surface region described. Since the model obtained is reproducible and predictive, both regions contain different conditions that should provide polymers in nanosized range.

### 2.3. Model Validation

For a mathematical analysis of experimental design data to be considered predictive, the model obtained must be further validated. Therefore, two conditions were chosen to validate the core-shell NFs formation to confirm the reproducibility of the model, considering fiber diameter as the response: the first replicated the condition that produced NFs with smallest diameters, 7.7% of PEO and 0.36% of NaCl; the second used 8.0% of PEO, without NaCl.

#### 2.3.1. Characterization of the Solutions for the Validation Experiments

The solution characterization of the best conditions chosen to validate the model was performed for both uniaxial and coaxial modes. The zeta potential of the uniaxial solutions, named V1 and V2 (with 7.7% of PEO and 0.36% of NaCl and with 8.0% of PEO without NaCl, respectively, both added with jussara pulp) was measured. The resulting values were −3.4 mV ± 1.1 and −2.2 mV ± 0.4, respectively, indicating a negative surface charge in both solutions. The absolute value of zeta potential can also be used as a measure of stability [28]. Therefore, the same analysis was applied to the coaxial solutions, carried out under identical conditions of PEO and NaCl concentrations as V1 and V2, but without jussara pulp, subsequently referred to as V3 and V4. The values of zeta potential for these solutions were −0.3 mV ± 0.2 for V3 and −0.4 mV ± 0.3 for V4. An increased zeta potential characterizes an enhanced stability of the solution, even if values are within a negative range [22]. Therefore, the addition of jussara pulp to polymeric PEO leads to greater stability of the structures.

#### 2.3.2. Characterization of the Nanofibers Obtained in the Validation Experiments

In this part of the study, uniaxial and coaxial modes were applied to elaborate core-shell NFs for both validation experiments to determine the ideal configuration and the operation conditions (Table 2) to produce the NFs. Up until now, all NFs have been generated using the uniaxial operational mode. For the coaxial process, the polymer solution (V3 and V4) and jussara pulp were injected separately into the equipment, and the concentrations of PEO, NaCl, and pulp used in this process are shown in Table 2. The flow rate applied in such setup was the only parameter altered from those employed in the uniaxial mode for shell and for the core was 600 and 200 (mL·h^−1^), respectively. Preliminary tests were accomplished to determine the flow rate applied (data not shown).

FE-SEM images from the uniaxial and coaxial electrospun fibers were once again used (Figure 1n–q) to measure their diameters (nm). Although samples did not present a consistent orientation, this time, a homogeneous fiber size was observed. Samples obtained by uniaxial (Figure 1n,o) and coaxial (Figure 1p,q) processes showed similarities in terms of fiber distribution and orientation, and there were no statistically significant differences in fiber diameters.

Samples V1 and V2 presented diameters of 112.6 ± 45 nm and 118.3 ± 64 nm (Table 2), respectively, and revealed no statistical difference between them. Indeed, these samples showed statistical equivalence to sample 2 from the CCDR, supporting the reproducibility of the model and further validating the experimental design. NFs produced through the coaxial conditions V3 and V4 displayed diameters of 121.1 ± 54 nm and 110.0 ± 47 nm, respectively (Table 2). Therefore, considering fiber size alone, the evidence herein indicates that any of the four conditions could successfully be used to fabricate jussara pulp NFs. Nevertheless, while a reduced size for biological systems is important for the food industry, the final product does not necessarily need to achieve diameters below 100 nm to be useful, as many compounds of interest have shown improvement in stability and function at diameter sizes above this reference, indeed ranging between 200 and 400 nm [12,13].

The thermal stability of the NFs synthetized for validation was evaluated using TGA, and the thermograms (TGA) derived thermogravimetry (DTG) curves for V1, V2, V3, and V4 are shown in Figure 3. TG analyses revealed a multistep weight loss curve for all samples. The first region at lower temperatures (<100 °C) originates from the loss of water [29] and solvent absorption. In contrast, the region at a higher temperature (>200 °C) is associated with decomposition of PEO. The first inflection is more evident in samples V1 (Figure 3a) and V3 (Figure 3c), indicating that more water was adsorbed by these samples possibly due to the salt present in their composition. For V1 (Figure 3a) and V3 (Figure 3c), weight loss occurs in two stages. For V2 (Figure 3b), this event occurs in three stages and, for V4 (Figure 3d), in four. The results of weight-loss ratios for all samples are summarized in Table 3.

Furthermore, it can be clearly seen that samples exhibit minimal weight loss up to 200 °C, when an exothermic peak appears, indicating that these NFs are stable at this temperature. Beyond that, all samples showed a total mass loss of approximately 80%. However, it must be note that sample V4 (Figure 3d) demonstrated better thermal stability since, at 250 °C, around 80% of the original sample was still preserved. Figure 3a,c shows that the position of the multistep weight loss curve shifts towards lower temperatures for V1 and V3 samples, respectively. These samples contain salt in their composition, which warrants their degradation at lower temperatures possibly due to the greater flexibility acquired by salt-bearing polymers. Such highly flexible polymer chains require less energy, and hence show diminished thermal resistance at lower temperatures. Thus, the presence of salt contributes to a decrease in the thermal stability of the polymer system [29].

Outlining thermal behaviors of NFs containing bioactive compounds is essential to promote their application in foodstuff as natural pigments. Remarkably, the most thermally stable ones increase their commercial interest for scaling-up process and industrial applications, since food production often require heating steps [29]. Lyophilized açaí fruit extract, possessing a high concentration of anthocyanins, presented an early degradation that started at approximately 100 °C, with maximum degradation at 162.5 °C [30]. These decomposition bands are visible only for the V4 sample (Figure 3e) in the region of the thermogram between 72 and 144 °C (thermal decomposition region of anthocyanins and açai extract, respectively), which is further supported the FTIR results. These evidence infer that the composition and the method for preparing the NFs directly influences their thermal stability, and that the saltless sample produced through a coaxial system presented higher thermal stability, since this method allows the bioactive compound to remain in the core of the structure and not on the surface as the uniaxial and, thus, revealed to be the more efficient manner to produce jussara pulp NFs.

To characterize the structure and composition of the NFs produced in both uniaxial and coaxial setups, FTIR, contact angle, and EDX experimental analysis were also carried out with the samples V2 and V4. Since no difference was detected in terms of diameter values among the four validatory samples, V2 and V4 were chosen for their simpler formulation, making for a more efficient process in terms of operation and economic resources.

The characteristic vibrational modes and wave numbers exhibited for these samples collected from the FTIR experimental spectra shown in Figure 4 are listed in Table 4, and they are in agreement with data reported in the literature. Figure 4 shows the FTIR spectra for PEO (8%), jussara pulp alone, sample V2 and sample V4. In the FTIR spectrum for the natural jussara pulp (Figure 4b), typical signals of the anthocyanin structure can be observed, referring to absorption bands at 1072, 1749, and 2923 cm^−1^, which are in agreement, respectively, to the bending vibration of C–O–C groups, the stretching vibration of C–O, and the saturated hydrocarbon groups (corresponding to methyl group) [31]. Moreover, bands related to a =C–O–C group of flavonoids and to the skeletal stretching vibration of the aromatic rings can be also detected (1072, 1506, and 1271 cm^−1^) [32]. The presence of the C-N group is assigned by bands between 1400 and 1450 cm^−1^ [33]. In addition, the existence of carboxylic acid, ester, or carbonyl groups can be observed at 1618 cm^−1^, with symmetrical and asymmetrical stretching vibration for the carboxyl ion (COO−) [34,35].

Figure 4c,d displays the FTIR spectra for samples V2 and V4, respectively. Small displacements were observed for bands referring to the PEO spectrum (Figure 4a) in both conditions, which is possibly due to interaction between the PEO and jussara pulp. However, the peaks at the 1250 cm^−1^ region, corresponding to the presence of ethereal oxygen and the crystal phase of PEO [36], appeared differently in spectra of samples containing jussara pulp. Similar effects were reported when PEO was used to produce NFs containing other bioactive compounds, such as cyclodextrin and β-carotene, when significant changes were detected in bands within the region between 1000 and 1400 cm^−1^ [37,38]. A peak in the 3370 cm^−1^ region was also identified herein, and that refers to the absorbance of water [39].

Analyzing the FTIR spectrum in the coaxial process (Figure 4d), it is possible to identify new bands and increased intensity and resolution of the PEO bands when compared to the jussara pulp spectrum. The new bands observed at 1413 cm^−1^ region reflect the presence of anthocyanins (C-N vibration) at 1066 and 1091 cm^−1^ (corresponding to the skeletal stretching vibration of the aromatic rings and =C–O–C group of flavonoids), at 1655 cm^−1^ (symmetrical and asymmetrical stretching vibration for the carboxyl ion (COO−)) and at 1780 cm^−1^ (stretching vibration of C–O). These results demonstrate a greater interaction between PEO and jussara pulp and imply that the coaxial method could successfully be used to produce nanofibers containing this fruit. Moreover, samples produced by the uniaxial process did not display the anthocyanins characteristics bands. This behavior is in agreement with data reported in the literature [21,40].

To investigated whether the electrospinning process affects wettability, the contact angle of the core-shell NFs was evaluated, and the assays were conducted with samples V2 and V4, both made up with PEO at 8%. Contact angles ranged from 37.4° ± 3.3° to 54.9° ± 2.9° showing no statistical difference (*p* < 0.01) among samples and, essentially, indicating that the structures possess hydrophilic character. PEO is commonly used in electrospun technique from grafting to blending, mainly due to its high hydrophilicity [41]. These results showed that the use of the jussara pulp with PEO matrix does not alter hydrophilicity of the final material, regardless of the method adopted for the process of NF formation.

Since the jussara pulp is rich in proteins containing N, EDX mapping (Figure 5) was carried out to monitor the elemental composition of the NFs in samples V2 (Figure 5a) and V4 (Figure 5b). A high accumulation in C and O counts is observed due to the composition of the polymeric PEO NF. A low amount of Na and Cl is also observed, possibly due to the presence of ions in the water used to solubilize the polymer. For V2, a small amount of N is observed, while in V4, the amount of N is higher. The presence of jussara pulp can be observed in both samples, revealing that uniaxial or coaxial setups can be used for the production of such NFs. However, these results suggest that the coaxial process was more effective in retaining greater amounts of jussara pulp in the polymer, as shown by the higher quantities of nitrogen atoms detected in the EDX spectrum of this sample, which are in agreement with those observed through FTIR and TGA analysis. In the uniaxial method of electrospinning, the jussara pulp is exposed for a very long period, especially when preparing the solution. Due to the natural instability of the bioactive compounds present in the pulp (such as anthocyanins), this long exposure can result in decreased pulp retention in the polymer. Similarly, Isik et al. [22] found that the uniaxial and coaxial electrospinning methods could be employed to encapsulate polyphenols using gelatin polymeric matrix; however, the coaxial systems showed higher efficiency, granting increased bioaccessibility of the compound. The characteristic absorption bands of the jussara pulp were preserved in the coaxial samples without significant changes, suggesting that this setup maintains most of the integrity of the pulp. Mazuco et al. [42] used a blend of maltodextrin and arabic gum to microencapsulated jussara pulp, and the association also proved to be a viable alternative for maintaining the integrity of the components. Other studies reported that biopolymers such as carboxymethyl, cyclodextrin and chitosan increased thermal stability of anthocyanins [43,44,45]. Herein, the present findings confirm that absence of salt further improved thermal stability of jussara pulp in PEO NFs.

This study validated the electrospinning process, operated either under a uniaxial or a coaxial mode, as a capable method to assimilate jussara pulp into core-shell NFs, while enhancing stability and, thus, suitability as an ingredient to be added to foodstuffs. However, the coaxial method allowed a greater amount of jussara pulp to be maintained to these structures. This is perceived as a positive feature in the present findings, taken that an increased availability of the compounds presents in the pulp for absorption by the gastrointestinal tract is obtained through a system that produces structures with higher stability and requires fewer handling steps—eliminating, for example, the stirring time necessary to produce the solutions for the uniaxial process, which can affect the pulp due to oxidation, light and temperature—and thus reducing possible issues related to degradation of bioactive compounds [7,21].

## 3. Materials and Methods

### 3.1. Jussara Sample

The jussara pulp was obtained from producers linked to the Jussara Project from Ubatuba City, São Paulo, Brazil. The frozen pulp was transported in coolers, freeze-dried (to preserve the bioactive compounds) and stored in a freezer until the analysis. Lyophilized jussara pulp was reconstituted in acetate buffer (pH 4.5) and filtered prior to the formulation of the solution applied in the uniaxial setup, while used unlinked in the coaxial configuration.

### 3.2. Solutions Preparation for Electrospinning

Different concentrations of PEO (900,000 g·moL^−1^, Sigma Aldrich, St. Louis, MO, USA) solutions, with and without NaCl, were prepared using jussara pulp reconstituted in a buffer as the solvent. PEO and NaCl concentrations used herein ranged from 6.0 to 8.0% (*w*/*v*) and 0.0 to 2.5% (*w*/*v*), respectively. The samples were homogenized in a magnetic stirrer and further used in the electrospinning process [21].

### 3.3. Electrospinning Process

Laboratory-scale electrospinning (FLUIDNATEK LE-10, BIOINICIA, Spain) equipped with a variable high-voltage 0–30 kV power supply was used to produce the NFs. The solutions were introduced in a 5 mL plastic syringe using a steel needle of 0.6 mm diameter under a flow rate of 3000 µL·h^−1^. Voltage and tip-to-collector distance (TCD) were fixed at 24 kV and 10 cm, respectively [21]. The samples were gathered from the collector, an anodized aluminum plate, at controlled room temperature (20–25 °C) and relative humidity (50–60%).

### 3.4. Experimental Design: Response Surface Methodology (RSM)

A Central Composite Rotatable Design (CCRD; 22 plus axial and central points) with three replicates at the central point, giving a total of 11 trials, was used as an attempt to maximize obtention of nanosized-scaled NFs. Diameters of NFs were considered as responses of the experimental design. Table 1 shows the values for the actual and coded levels used in the CCRD with their respective diameter sizes. An estimate of the effects was obtained by evaluating the differences in process performance caused by a change from the low (−1) to the high (+1) levels of the corresponding variable (PEO and NaCl). After analyzing the RSM results, the best conditions for the production of NFs were determined, and the model was validated in triplicate. All the experimental design samples were produced using uniaxial electrospinning (Figure 1a).

### 3.5. Electrospinning Validation

To validate the experimental design model, two conditions were chosen based on the best results obtained. The solutions to produce the validatory NFs were prepared under the following conditions: V1 and V3 comprised 7.7% (*w*/*v*) of PEO and 0.36% (*w*/*v*) of NaCl; and V2 and V4 were loaded with 8% (*w*/*v*) of PEO and 0% NaCl. While V1 and V2 designate the samples obtained through uniaxial electrospinning, V3 and V4 depict those samples produced through a coaxial setup. For the uniaxial mode, PEO and the NaCl were solubilized into jussara pulp, whereas, for the coaxial setup, the reagents were solubilized in acetate buffer (pH 4.5). The parameters of the electrospinning validation process were the same as described before (voltage of 24 kV and TCD of 10 cm). The solutions were introduced in a 5 mL plastic syringe and a steel needle of 0.6 mm diameter and the flow rate was at 3000 µL·h^−1^ (Figure 1a) were used for uniaxial operation. For the coaxial system, in turn, two concentric steel needles of 1.4 mm and 0.6 mm inner diameters were used for polymer solution and jussara pulp, respectively (Figure 1m). Flow rates for the PEO solution was at 600 µL·h^−1^, while jussara pulp flowed out at 200 µL·h^−1^. The applied voltage and TCD parameters were maintained, and samples were collected at controlled room temperature (20–25 °C) and relative humidity (50–60%).

### 3.6. Characterization of the Nanofibers from Experimental Design

#### FE-SEM Images and EDX Analysis

Field emission scanning electron microscopy (FE-SEM Supra 35 VP-equipment, Carl Zeiss, Germany) and Energy-dispersive X-ray spectroscopy (EDX) were used to obtain micrography images of the samples. DiameterJ (plugin for the ImageJ software) assisted analysis was used to expedite measurement in FE-SEM images and reduce bias from manual data processing [46].

### 3.7. Characterization of the Solutions for Validation

The zeta potential was measured by a dynamic light scattering instrument, a Zetasizer (Malvern Instruments, Malvern, United Kingdom) with an MPT-2 titrator. The solutions were dispersed into 0.1% (*w*/*v*) ethanol. The viscosity of the solutions prepared according to the conditions presented in Table 1 was measured with a Reometer model MCR 92 (Anton Paar, Graz, Austria). The samples were further submitted to conductivity and pH analysis by means of measurements obtained with a digital conductivimeter (Hanna HI 8733, São Paulo, Brazil) and a pHmeter (Hanna HI5522, São Paulo, Brazil), respectively.

### 3.8. Characterization of the Nanofibers for Validation

The NFs were characterized by their FE-SEM images and by the micrographs obtained thereof, from which their diameter sizes were extracted using measurements handled by the tool DiameterJ) [46]. Additionally, Fourier-transform infrared spectroscopy (FTIR) (Bruker Alpha-P, in the 4000–500 cm^−1^ range) was used to provide the characteristic fundamental vibrational modes and wavenumbers from experimental spectra.

Hydrophobicity was determined by measurement of the NF’s surface contact angle using a sessile drop method in a Rame-Hart goniometer (Model 260-F, Washington, DC, USA) coupled to the software DROPimage Advanced. Deionized water was used as the wetting liquid and the droplet volume was fixed at 5 μL for each standard wetting liquid. Under room temperature (26 ± 1 °C), this parameter was established as the mean and standard deviation of 10 assessments at random locations on the surface of the samples. Samples presenting surface contact angles bellow 90° were considered hydrophilic, while those above 90° were designated as hydrophobic.

Thermal stability of the core-shell NFs was characterized by thermogravimetric analysis using a TA Instruments Q-50 apparatus (Mettler-Toledo, Barueri-SP, Brazil), under a temperature range of 0–700 °C and an N2 atmosphere with a scan rate of 10 °C·min^−1^. Moreover, to confirm the composition of the formed structures, samples were also mapped by EDX spectroscopy with an FE-SEM Philip XL-30 TMP coupled to an Oxford EDS.

### 3.9. Statistical Analysis

Analysis of variance (ANOVA) was used to evaluate the results for the experimental design (level of 95% confidence, *p* < 0.05), which was carried out using Fisher’s statistical test. The F-value calculated therein indicates the influence (significance) of each factor controlled in the model tested [47]. The measurements from the assays were carried out independently in triplicate and compared by applying analysis of variance (ANOVA) using the degree of significance of 95% (*p* < 0.05), with Tukey’s post hoc test. Statistica 13.0 software was used to analyze the results.

## 4. Conclusions

Electrospinning provided nanosized structures containing jussara pulp, indicating, therefore, that it was a successful method for producing core-shell NFs included with the bioactive compound. A mathematical model was obtained by an Experimental Design from PEO and NaCl concentration variables, enabling further solution outsets addressed at inversely proportional quantities of this substances in electrospinning solutions. In addition, it was proved that the presence of the salt does not interfere with the size of nanofibers once elevated concentrations of PEO are used. The parameters employed herein disclosed that a coaxial mode of operation provided improved nanocarriers for jussara pulp, as this setup preserved the integrity of the pulp, delivered thermal stability and qualitatively enhanced the presence of the pulp in the core-shell NFs. Electrospinning provided nanometric structures containing jussara pulp, thus proving to be a successful method for producing core-shell NFs included with the bioactive compound. Therefore, the use of PEO through the core-shell nanofiber formation method is a promising way to maintain the integrity of jussara pulp, and consequently the stability of bioactive compounds, enabling its industrial application as a natural pigment in the processing of foodstuffs and beverages.

## Figures and Tables

**Figure 1 molecules-26-01206-f001:**
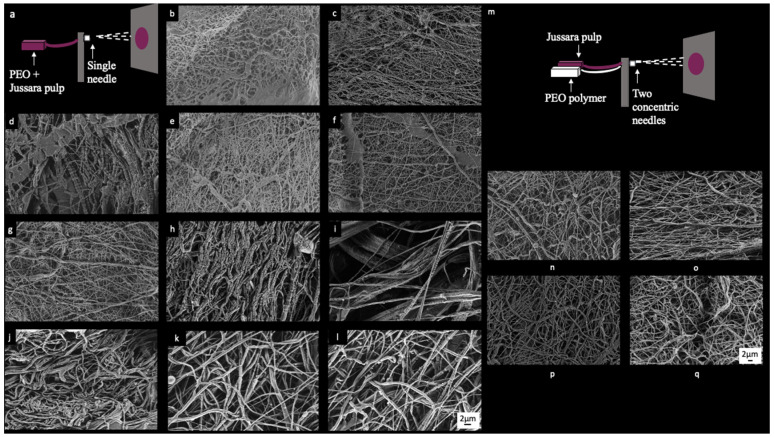
FE-SEM images electrospinning samples: (**a**) schematic view of uniaxial electrospinning, (**b**) sample 1 (6.3% PEO and 0.36% NaCl), (**c**) sample 2 (7.7% PEO and 0.36% NaCl), (**d**) sample 3 (6.3% PEO and 2.14% NaCl), (**e**) sample 4 (7.7% PEO and 2.14% NaCl), (**f**) samples 5 (6.0% PEO and 1.25% NaCl), (**g**) sample 6 (8.0% PEO and 1.25% NaCl), (**h**) sample 7 (7.0% PEO and 1.25% NaCl), (**i**) samples 8 (7.0% PEO and 1.25% NaCl), (**j**) sample 9 (7.0% PEO and 1.25% NaCl), (**k**) sample 10 (7.0% PEO and 1.25% NaCl), (**l**) sample 11 (7.0% PEO and 1.25% NaCl), (**m**) schematic view of coaxial electrospinning, (**n**) sample V1 (7.7% of PEO and 0.36% of NaCl), (**o**) sample V2 (8.0% of PEO), (**p**) sample V3 (7.7% of PEO and 0.36% of NaCl), and (**q**) sample V4 (8.0% of PEO).

**Figure 2 molecules-26-01206-f002:**
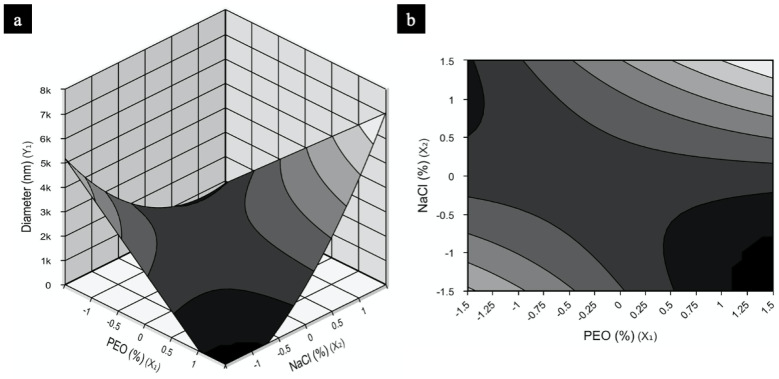
PEO response and contour surfaces for fiber diameter containing as a function of concentrations PEO (%) and NaCl (%): (**a**) response surface, (**b**) contour surface.

**Figure 3 molecules-26-01206-f003:**
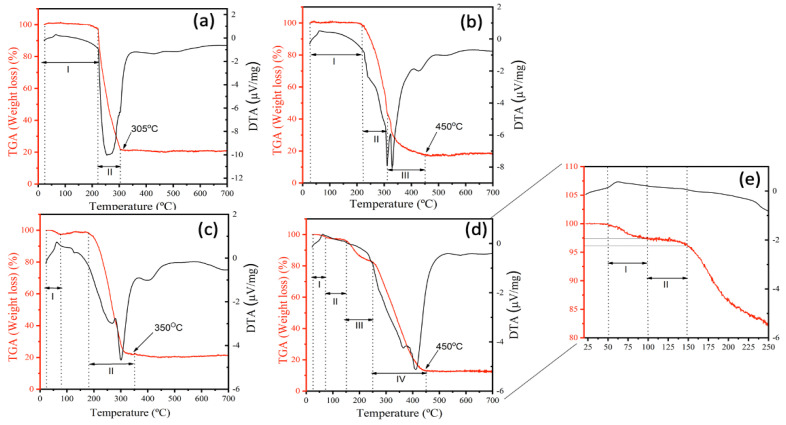
TGA thermograms of the samples for validation obtained from the V1 sample (**a**), the V2 sample (**b**), the V3 sample (**c**), the V4 sample (**d**) and the zoom from 25 to 250 °C (**e**).

**Figure 4 molecules-26-01206-f004:**
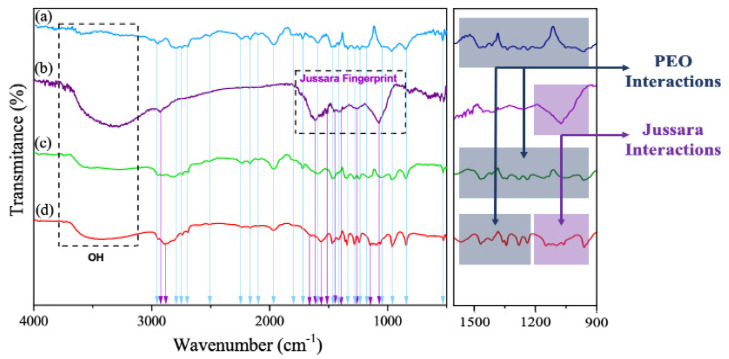
FTIR spectra: (**a**) PEO 8% sample, (**b**) jussara pulp, (**c**) PEO 8% and jussara sample (V2), and (**d**) PEO 8% and jussara sample (V4), at room temperature.

**Figure 5 molecules-26-01206-f005:**
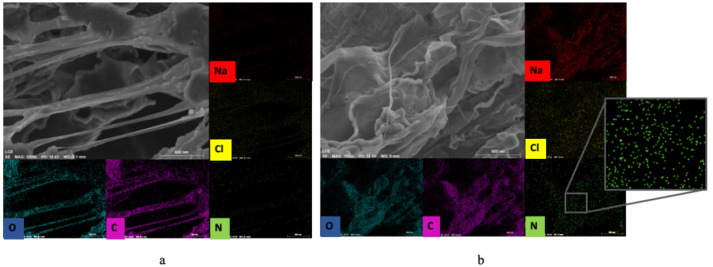
EDX map from the conditions of validation resulting in the option of samples: (**a**) sample V2, 7.7% of PEO and 0.36% of NaCl; (**b**) sample V4, 8.0% of PEO.

**Table 1 molecules-26-01206-t001:** Coded and experimental values and the diameter measurements (nm) of core-shell nanofibers with jussara pulp.

Sample (*)	PEO (%)	NaCl (%)	Diameter (nm)
1	−1 (6.3)	−1 (0.36)	2454.0
2	1 (7.7)	−1 (0.36)	135.0
3	−1 (6.3)	1 (2.14)	1131.0
4	1 (7.7)	1 (2.14)	4268.0
5	−1.41 (6.0)	0 (1.25)	1647.0
6	1.41 (8.0)	0 (1.25)	2076.0
7	0 (7.0)	−1.41 (0.00)	2287.0
8	0 (7.0)	1.41 (2.50)	3847.0
9	0 (7.0)	0 (1.25)	1513.0
10	0 (7.0)	0 (1.25)	1480.0
11	0 (7.0)	0 (1.25)	1475.0

* All samples were generated using uniaxial electrospinning (R^2^ = 0.94).

**Table 2 molecules-26-01206-t002:** Operation mode and process conditions for validation assays and the diameter measurements (nm) of core-shell NFs.

Electrospinning	Sample	PEO (%) (*w*/*v*)	NaCl (%) (*w*/*v*)	Jussara (%)	Feeding Rate (μL·h^−1^)		Diameter (nm)
					PEO	Jussara	
Uniaxial	V1	7.7	0.36	5	150		112.6 ± 45^a^
Uniaxial	V2	8.0	0	5	150		118.3 ± 64^a^
Coaxial	V3	7.7	0.36	5	600	200	121.1 ± 54^a^
Coaxial	V4	8.0	0	5	600	200	110.0 ± 47^a^

Diameter values, expressed as mean ± SD, equal letters in the same column represent no statistically significant difference (*p* < 0.05).

**Table 3 molecules-26-01206-t003:** Results of weight-loss ratios for the samples from the validation experiments.

Sample	Stage I T (^o^C)	Weight Loss (%)	Stage II T (^o^C)	Weight Loss (%)	Stage III T (^o^C)	Weight Loss (%)	Stage IV T (^o^C)	Weight Loss (%)	Total Weight Loss (%)	Maximum Temperatures of Weight Loss (Tm)
V1	24–221	3	221–305	76					79	305
V2	27–217	1	217–312	57	312–450	24			82	450
V3	22–76	3	180–350	76					79	350
V4	25–72	1	72–144	3	144–247	15	247–450	70	89	450

**Table 4 molecules-26-01206-t004:** Wavenumbers and assignments of IR bands exhibited by samples V2 and V4. Where ν = stretch, δ = scissor/deformation, ω = wag, ρ = rock, τ = twist. The subscripts s and as refer to symmetric and asymmetric vibrational modes, respectively.

Wavenumber (cm^−1^)	Assignment Sample V2	Assignment Sample V4
531	*τ* (C-C)	*τ* (C-C)
850	ρ (CH_2_) + δ (C-O-C)	ρ (CH_2_) + δ (C-O-C)
966	ρ_as_ (CH_2_) + ν (CH_2_)	ρ_as_ (CH_2_) + ν (CH_2_)
1053	ν (C-C) + ρ (CH_2_)	ν (C-C) + ρ (CH_2_)
1178	ν_s_ (C-O-C)	ν_s_ (C-O-C)
1235, 1283	τ_s_ (CH_2_), τ_as_ (CH_2_)	τ_s_ (CH_2_), τ_as_ (CH_2_)
1328, 1383	A doublet ω (CH_2_)	A doublet ω (CH_2_)
1456, 1591	δ_s_ (CH_2_), δ_as_ (CH_2_)	δ_s_ (CH_2_), δ_as_ (CH_2_)
1726 -2015	ν (C=O)	ν (C=O)

## Data Availability

The data presented in this study are available in Appendix A.

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
