# Peer review of "Uniaxial and Coaxial Electrospinning for Tailoring Jussara Pulp Nanofibers"

_molecules, 2021, doi:10.3390/molecules26051206_

Round 1
Reviewer 1 Report
This manuscript compared uniaxial and coaxial electrospinning operation modes for the production of core-shell jussara pulp nanofibers (NFs). The topic is significance and worthy of studying. The obtained results indicated an optimized approach for the preparation of NFs. The manuscript is well written and well organized. I suggest to accept after addressing the following points:
1. The underlying mechanisms of the differences between uniaxial and coaxial electrospinning operation modes should be explained in more detail together with the obtained results.
2. The differences in responsive capability of NFs obtained by uniaxial and coaxial electrospinning operation modes should be compared.
3. How the parameter of electrospinning operation influence the structure of NFs should be explained in more detail.
4. The potential applications of NFs should be proposed and emphasized in the conclusion section.
Author Response
To: Molecules
Manuscript ID: molecules-1108333
Title: Uniaxial and coaxial electrospinning for tailoring jussara pulp nanofibers
Dear Editor,
Thank you for the reviewer’s useful comments and suggestions for our manuscript. We have modified the manuscript according to the reviewers’ recommendations, and the detailed point by point corrections are listed below. The modifications requested by each reviewer were significant to improve our work. All sentences/words listed below are highlighted in the revised version of the manuscript.
Reviewer #1: This manuscript compared uniaxial and coaxial electrospinning operation modes for the production of core-shell jussara pulp nanofibers (NFs). The topic is significance and worthy of studying. The obtained results indicated an optimized approach for the preparation of NFs. The manuscript is well written and well organized. I suggest to accept after addressing the following points:
Response: The authors acknowledge the nice comments of the reviewer and the time spent evaluating this work.
1. The underlying mechanisms of the differences between uniaxial and coaxial electrospinning operation modes should be explained in more detail together with the obtained results.
Response: The alteration requested by the referee was done.
2. The differences in responsive capability of NFs obtained by uniaxial and coaxial electrospinning operation modes should be compared.
Response: The comparation was added as required.
3. How the parameter of electrospinning operation influence the structure of NFs should be explained in more detail.
Response: The information was added as required.
4. The potential applications of NFs should be proposed and emphasized in the conclusion section.
Response: The information was added in the conclusion section.
Reviewer 2 Report
and The Manuscript titled" Uniaxial and coaxial electrospinning for tailoring jussara pulp nanofibers" is very interesting study on utilisation of jussara pulp for fabrication of nanofibers using electrospinning technique.
The article is of interest of the reader of the Molecules journal and it should be considered for publication after addressing few comments:
- The abstract seems to be loosely connected to the substance of the article and the real aim, same is with the explanation of the aim within the background of the study.
- Please change the expression "uniaxial electrospinning" for electrospinning, since it is a most standard and typical way of this process and there is no need to obfuscate the process by giving it a fancy name.
- FTIR results: there is a overinterpretation/misinterpretation of the results regarding some new bands and great interaction between jussara and PEO. First of all the spectra in indicates region show something that should not be considered as peaks; second if a coaxial electrospinning is applied and one of the electrospinning material is in a core of resulting core-shell fibres, and the fibres are obtained under optimised conditions so the shell material is continous, the internal material should not be present on the surface of the fibres.
- Figure 2 should be improved, the morphology of the fibres is barely visible on the images - I suggest making the images larger.
Author Response
To: Molecules
Manuscript ID: molecules-1108333
Title: Uniaxial and coaxial electrospinning for tailoring jussara pulp nanofibers
Dear Editor,
Thank you for the reviewer’s useful comments and suggestions for our manuscript. We have modified the manuscript according to the reviewers’ recommendations, and the detailed point by point corrections are listed below. The modifications requested by each reviewer were significant to improve our work. All sentences/words listed below are highlighted in the revised version of the manuscript.
Reviewer #2: The Manuscript titled" Uniaxial and coaxial electrospinning for tailoring jussara pulp nanofibers" is very interesting study on utilisation of jussara pulp for fabrication of nanofibers using electrospinning technique. The article is of interest of the reader of the Molecules journal and it should be considered for publication after addressing few comments:
Response: The authors thank the reviewer nice comments and the time spent evaluating the work.
- The abstract seems to be loosely connected to the substance of the article and the real aim, same is with the explanation of the aim within the background of the study.
Response: The abstract was changed as requested.
- Please change the expression "uniaxial electrospinning" for electrospinning, since it is a most standard and typical way of this process and there is no need to obfuscate the process by giving it a fancy name.
Response: All the consulted articles used to discuss the data presented in the present work the specified the electrospinning operational mode, uniaxial or coaxial. The differentiation is important to avoid a misleading interpretation of the article readers, especially in the present work since there is a comparation among the results obtained from both uniaxial and coaxial electrospinning.
- FTIR results: there is a overinterpretation/misinterpretation of the results regarding some new bands and great interaction between jussara and PEO. First of all the spectra in indicates region show something that should not be considered as peaks; second if a coaxial electrospinning is applied and one of the electrospinning material is in a core of resulting core-shell fibres, and the fibres are obtained under optimised conditions so the shell material is continous, the internal material should not be present on the surface of the fibres.
Response: In this work the FTIR technique was used to characterize the structure and composition of NFs produced in uniaxial and coaxial configurations. It was possible to verify, in V4 samples, bands in the region between 900 and 1200 cm-1 (jussara fingerprint), that are not visible in the PEO and V2 spectra. This result shows that there was an interaction between PEO and Jussara Pulp in the V4 sample. The FTIR technique is widely used to analyze a composition and the core-shell structure of materials. The behavior observed in this work has already been reported in the literature by similar works. These references were inserted in the work.
- Figure 1 should be improved, the morphology of the fibres is barely visible on the images - I suggest making the images larger.
Response: The required alteration was done.